# Solo Travel Research and Its Gender Perspective: A Critical Bibliometric Review

Almudena Otegui-Carles, Noelia Araújo-Vila * and Jose A. Fraiz-Brea

Business and Tourism Faculty, Universidade de Vigo, 32004 Ourense, Spain
* Correspondence: naraujo@uvigo.es

**Abstract:** Solo travel continues to be an under-researched area in the field of tourism, hospitality, and events. After the COVID-19 pandemic, it has become necessary to review the knowledge acquired so far. In addition, the 2030 Agenda calls for more studies to understand the relationship between gender and tourism. Because of these facts, and with the aim of analyzing the progress and gaps in academic publications on solo travel in recent years, a bibliometric and content analysis review of the existing scientific literature on solo travel published in Scopus, ProQuest, and the Web of Science in the last 5 years was carried out, focusing the analysis on the gender perspective applied to these investigations. The results showed that research focused on solo travel should increase; this research should segment solo travelers, and comparisons should be realized between those segments and with other tourists who travel accompanied. To do so, a consensual definition of solo travelers is necessary. In addition, research should be extended to other regions and expand the field of analysis beyond motivations, experiences, or constraints. Research focused on solo female travelers should continue because while women cannot travel under the same conditions as men effective gender equality cannot be achieved.

**Keywords:** solo travel; bibliometric review; gender equality; female; risks; COVID-19; Scopus; Web of Science; ProQuest; sustainable development goals

## 1. Introduction

Social distancing has made solo travel more desirable [1,2], thereby receiving a boost since the COVID-19 pandemic began [2,3]. Nevertheless, solo travel had already become one of the fastest growing types of tourism in recent years [4–7], assuming a significant contribution to the travel market [8], estimated to account for 18% of total travel bookings [9,10]. The increase in the number of people traveling alone is due, on the one hand, to structural changes in society [11], such as delays in the marriageable age or even an increase in the unmarried; an increase in the number of people living alone [5,12]; and increased consumption, individualism, or fewer children [5]. These lifestyle changes have particularly influenced women, who have passed from being relegated to household chores to joining the workforce and attaining economic empowerment [13]. Having increasingly individualistic lifestyles has led to an increase in women choosing to travel alone [14], even more than men [9]. On the other hand, the COVID-19 pandemic led not only to solo travel becoming more appealing for maintaining social distancing but also unintentionally caused people to socialize less and less [15].

The study and understanding of this market is necessary and important [4], not only because of its economic and managerial implications but also because of its social implications. Solo travel is strongly involved in the achievement of Goal 5 of the 2030 Agenda. Since the growth of this type of tourism is more pronounced among women [13], its study implicitly includes the analysis of associated gender factors, such as safety, sexual harassment, and its contribution to gender equality. For these reasons, it is important to broaden the knowledge of this type of tourism.

An indication of the growing interest in solo travel is an increase in searches for the term "Solo Travel" on Google in recent years, with an increase of 500% since 2009 and a higher search volume in Asia [16]. In 2019, searches for the term "Solo Women Travel" increased by 203% compared to 2018 [9]. Similarly, Pinterest reported a 350% increase in posts tagged as "Solo Female Travel" [9]. In Figure 1, it is possible to see the interest over time in "Solo Travel" Google searches since 2017 [17]. The interest has been growing since 2017, reaching its peak in December 2019 and its lowest point in May 2020 in the midst of the COVID-19 pandemic; however, interest is rebounding, and in May 2022, it was close to reaching the levels of 2019.

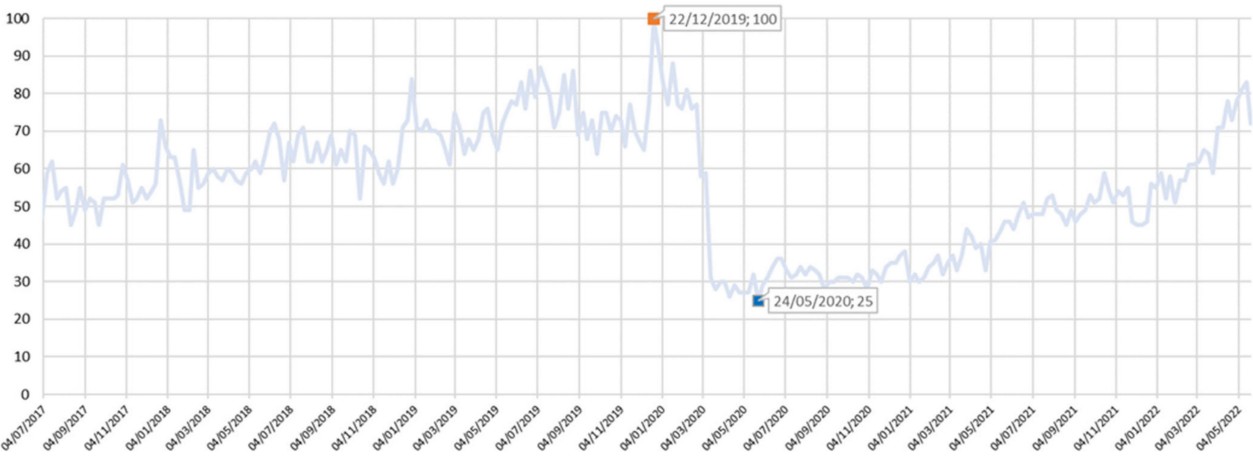

**Figure 1.** Interest over time in solo travel Google queries.

Despite this, solo travel continues to be an under-researched area in the field of tourism, hospitality, and events [5,8,11,12,14,18,19]. Furthermore, research on solo travel is focused on solo female travel [2,18], neglects the experiences of men [20], and focuses on solo travelers from specific regions [2,6,18,21]. There is also little research on solo travel that compares the motivations of men and women [6].

There is no consensus to establish a definition of solo travel [6,12]; however, there is an agreement that two types of solo travelers exist: those traveling alone "by default" and those who do it "by choice" [6,12,19,20]. A lack of company could be a reason for traveling alone [6,22], but many of the people who travel alone do have someone to travel with, so traveling alone is a choice of their own [8,10]; even having someone to travel with, they might decide and prefer to travel alone [11]. Instead of traveling with people with different tastes or different social values, some people prefer to travel alone [13], and there are even people who decide to travel alone simply because they "hate people" [6]. There are also people who travel alone for the first time "by default" because they do not have travel companions, but after the first experience, they became solo travelers "by choice" [6]. The discussion is whether to define solo travelers as those who arrive alone at a destination and remain alone throughout the trip or as those who arrive alone at a destination and join a group or an organized tour [6,10–12,15,20]. Laesser et al., (2009) established a broad set of solo travelers of four types: people living alone who travel alone; people living alone who join a group or an organized trip; people who do not live alone and travel alone; people who do not live alone and travel alone but join a group or an organized trip.

Solo travelers are a heterogeneous group [1]; therefore, they have very different travel motivations. Some of these motivations are not specific to solo travelers, but to travelers in general [6], although, in general, the attitudes of solo travelers are different from travelers who travel accompanied [23]. These motivations may differ if it is the first time traveling alone [18] or if it is an experienced solo traveler. Academic publications made so far indicate that, for women who decide to travel alone, the objective is not the trip itself, but the experience [14] that allows them to leave their comfort zone and

feel independent and autonomous [11,14,21,24]. Women traveling alone seek personal growth and development [10,14,21] by increasing self-confidence and making themselves feel empowered [8,10,12,25,26]; having sex, gaining prestige [2], or escaping their family roles [8,10,21,25] are other motivations. The motivation of escaping is especially important for Muslim women, who feel an "exciting sense of emancipation" [27] when they distance themselves from imposed norms. Women also seek freedom [10,12,21,25,26]; meet people and new cultures [8,10,12,26]; do new and authentic things [26]; experience adventures [21]; expand their visions of the world [24]; and increase their knowledge [14]. Traveling alone allows women to increase their well-being, happiness [28], and relaxation [8,25], which, in turn, helps them to overcome stress, sadness, and depression [8].

Few studies have investigated the differences between genders when traveling alone [5,10,11,20]. These few studies suggest that men have different preferences than women [19,26]: they seek contact with nature, explore new places, try new foods, meet people from other cultures, travel to destinations marked by poverty, and even seek danger [20]. They like adventure [26], local arts, and cultural activities [29]. Young, single males traveling alone can also look for sports and party holidays [20]. These findings suggest that, for men, traveling alone does not have the same component of meaningful travel regarding personal feelings as it does for women.

There are also not many studies comparing solo travelers with accompanied ones. A few studies indicate that, compared to tourists who do not travel alone, solo tourists consider the opinions or concerns of their families and friends about their travel decisions [18,30], especially the first time they travel alone [18]. Some studies indicate that solo travelers had fewer complaints than people who travel accompanied [31]. Other studies, however, indicate that solo travelers have higher requirements for indoor environmental quality, indoor air quality, acoustic environments, and playful environments [32] and that they have a less satisfying travel experience than those who travel accompanied [33]. There are also contradictions in terms of the duration of the trip; there are investigations that indicate that accompanied tourists tend to stay in the destination longer than solo travelers [15]; however, other data point to the fact that solo travelers make longer trips than other tourists, with about 19 days on average [9].

Diseases and injuries are more common for solo travelers compared to those who travel with companions [34]. People who travel alone, regardless of their gender, are more vulnerable [10–12] and have a greater sense of insecurity and risk perception [2], that can decrease after a first trip alone [18]. In this regard, there are also contradictions. Sung et al., (2021) stated that accompanied tourists are more sensitive to the travel risks of the COVID-19 pandemic compared with solo travelers, and Bačík et al., (2020) stated that people traveling with family or friends care more about safety and security than solo travelers. In addition, other health-related academic studies, which have studied the behavior of tourists with respect to prevention and control health practices with some relevant results [35], contradict previous tourism academic studies pointing to the fact that solo travelers have fewer concerns about health than people that travel accompanied. Thus, for example, Khoury et al., (2021) stated that solo travelers are less adherent to taking antimalarial chemoprophylaxis when traveling to endemic areas compared with those who travel with a family member. Despite these possible contradictions, the truth is, without anyone to accompany them on the journey who can provide them assistance, people who travel alone are more exposed to diseases, crime [11], language barriers, and natural disasters [30], and even more likely to get lost [10,36]. Both genders may also be affected, at some point, by a sense of loneliness [10,12] and even boredom [25], as well as by the negative attitudes of local inhabitants, stigmas, or less friendly service than people who travel accompanied [18,21]. This less friendly service is produced by a lack of entertainment activities created for solo travelers [11] and by a lack of options for people traveling alone; on the contrary, in many hotels, there is an obligation to pay a single room supplement [10,11,16].

Women are particularly affected by the risks of traveling alone; they are, therefore, more concerned with safety issues than men [24,30], which is one more manifestation of male hegemony [30,36]. That is why the study of solo travel involves the study of gender inequalities. Women who travel alone feel more vulnerable, especially at night [37], avoiding going out to dinner alone in the evening [1,11,20]. This is because they risk unwanted male attention, sexual harassment, and even rape [7,21,24,26,36,38] or appear sexually available for the simple fact of being alone [16]. This leads women who travel alone to plan the trip with the aim of avoiding dangerous encounters [20] and to feel intimidation when they pass through places where they feel more vulnerable [26]. This reduces their enjoyment and interactions, and even leads them to modify their way of dressing [26]. Women are afraid of the threat of male violence even in hotel accommodation and transportation, considering elevators and hotel corridors dangerous spaces [39].

Depending on the culture and religion, some women are especially vulnerable such as, for example, Asian women [26,36,40]. Family and social customs continue to have a very deep influence in some societies [27], and there is still much stigma in some cultures about women living or traveling alone [16,24]. They may have more difficulty traveling; such is the case of Iranian women who must have the permission of their fathers or husbands to travel [21,24], or Muslim women who might be forced to travel accompanied by what is known in Shariah as a "Mahram" [28]. Not to mention the fact that host countries can stereotype other religions [28], and traveling wearing the hijab can create uncomfortable situations with Islamophobic people [27], which leads them to wear a hat instead of a hijab to avoid this stigma [28]. The difficulties in obtaining a visa for most Muslim citizens [27], and the fact that traveling around their own country Asian women can be confused for "sex workers" [36], are all added difficulties, and women suffering these difficulties experience a form of double discrimination [40] for being women and being from certain countries. However, studies indicate that the additional barriers and risks faced by women traveling alone do not stop their travel plans [7,27,30]. An example of this is the fact that India is one of the destinations for single female travelers that has grown the most, despite being a dangerous destination for women [7,8]. In turn, the number of Muslim, Indian, and other Asian women who travel alone has not stopped growing [13,28] despite the added difficulties these women encounter. Furthermore, some studies suggest that confronting stereotypes, gender roles, and insecurity confers an added reward to the journey [8], generating a feeling of empowerment in them [36].

In Figure 2, it is possible to see interest by region in the queries made on Google using the term "Solo Travel" [17], with 100 being the maximum interest. The highest interest comes from Asian countries, in line with the previous literature.

Since the COVID-19 pandemic, it has become necessary to review the current knowledge and to understand the changes that have occurred in tourism, as is the case of solo travel, which is growing in strength. In addition, the 2030 Agenda and the study of sustainable development goals (SDGs) "*call for more studies that offer alternative ways of understanding the relationship between gender and tourism development and provide inspiration for creative and progressive ways of harnessing tourism to meet this goal*" [27] (p. 2). Tourism offers opportunities to achieve the SDG of gender equality; however, as presented in the previous literature and as other studies indicate, there are still large inequalities [21,41] that have even increased since the COVID-19 pandemic [42–44]. Tourism transforms gender roles and stereotypes [27], and solo travel tourism offers women the opportunity to confront patriarchal systems and the norms imposed by those systems [21,27]. Because of these facts, and with the aim of analyzing the progress and gaps in academic publications on solo travel in recent years, establishing a starting point for new research, relating it to gender equality, and expanding scientific literature related to solo travel and gender equality, both fields that are still rather unexplored, this bibliometric review of the existing scientific literature is presented. To this end, a bibliometric and content analysis review of the existing scientific literature on solo travel published in Scopus, ProQuest, and the Web of Science in the last

5 years was conducted, focusing the analysis on the gender perspective applied in these investigations.

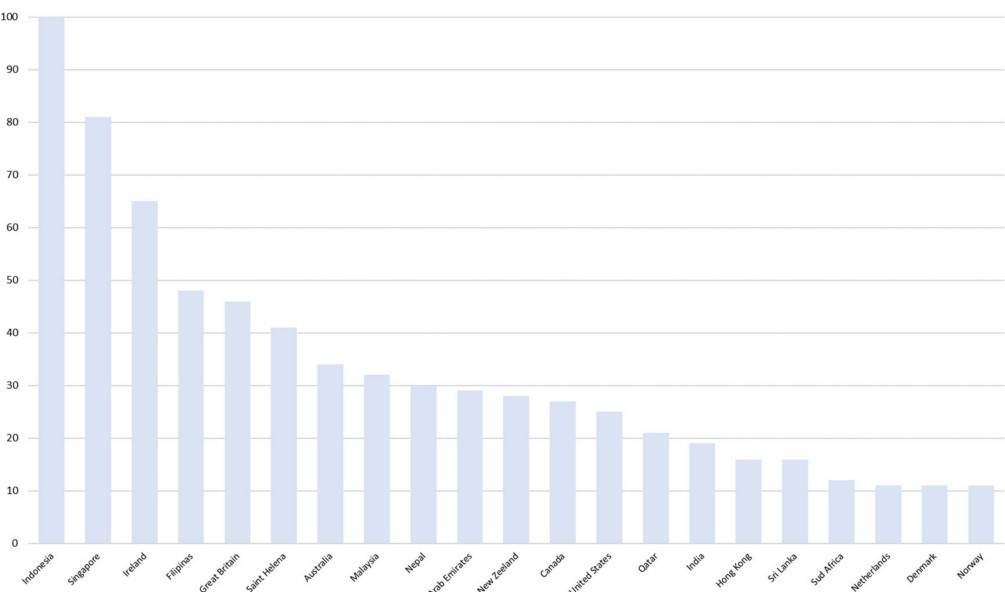

**Figure 2.** Interest by region in solo travel Google queries.

## 2. Materials and Methods

The objective of a bibliometric review is to summarize the available primary research [45] accurately and reliably [46] to establish the state of existing knowledge [45]. Authors have used the same technique in other research, showing it to be a technique that is successfully applicable and with easily understandable results. Following the PRISMA Statement [47], the PRISMA 2020 checklist was completed, and a flow diagram is included in this article (Figure 3). This systematic review includes two main techniques: evaluative methods and relational methods [48,49]. Authors decided to use these techniques because, as noted by Koseoglu et al., (2016): "the bibliometric relational techniques explore relationships among the research fields, the emergence of new research themes and methods, or co-citation and co-authorship pattern" (p. 182). The co-word structure is a content analysis technique that analyses word frequency in a text and seeks to find patterns to build concepts in a certain area [49,50]. The specific type of content analysis employed in the present work was categorical content analysis, which consists of dismembering the texts into units, or categories, according to pre-established criteria [51]. According to Molinos et al., (2016) [52], this kind of analysis is a powerful tool to detect key themes in scientific articles.

The data collection procedures were carried out in July 2022, which consisted of a search of the words "Solo near/1 travel*" in titles, abstracts, and keywords in the Scopus, Web of Science, and Proquest (scientific journals) databases in the last 5 years, which is the lifespan of a manuscript in social sciences [53]. Therefore, the search was from 2017 to 2021; in addition, articles published up to 10 July 2022 were also included. The Scopus and Web of Science databases were chosen because they are two of the major multidisciplinary databases that exist [54], with the largest set of citation data and high-quality abstracts of peer-reviewed literature [55]. ProQuest Central consists of independent databases with full search capabilities, providing access to 47 complete ProQuest databases with a variety of content types in more than 175 subjects [56].

As can be seen in the PRISMA 2020 flow diagram (Figure 3), 161 records were found: 82 from Scopus, 60 from Web of Science, and 19 from ProQuest. In total, 1 record was removed from ProQuest because it was duplicated; 27 were removed from Scopus; and 9 were removed from Web of Science because they were not articles. After that, another

18 records were removed because they were not related to the subject of the study; most of them referred to solo transport technologies. In total, 107 articles remained eligible; 43 were excluded from Web of Science because they were both in Scopus and Web of Sciences, and 9 were excluded from Proquest because they were both in Scopus and Proquest. Finally, 55 articles were included for review. The bibliographic information of the articles included for review was organized, classified, and summarized. Then, word frequency analysis and an analysis of key topics were applied to explore the contents and relationships between the research topic and its associated topics. Word frequency analysis and the content analysis method were used to quantify qualitative data [57]. For a better understanding of the data obtained, these have been presented with the support of various tables and figures along with their respective explanations.

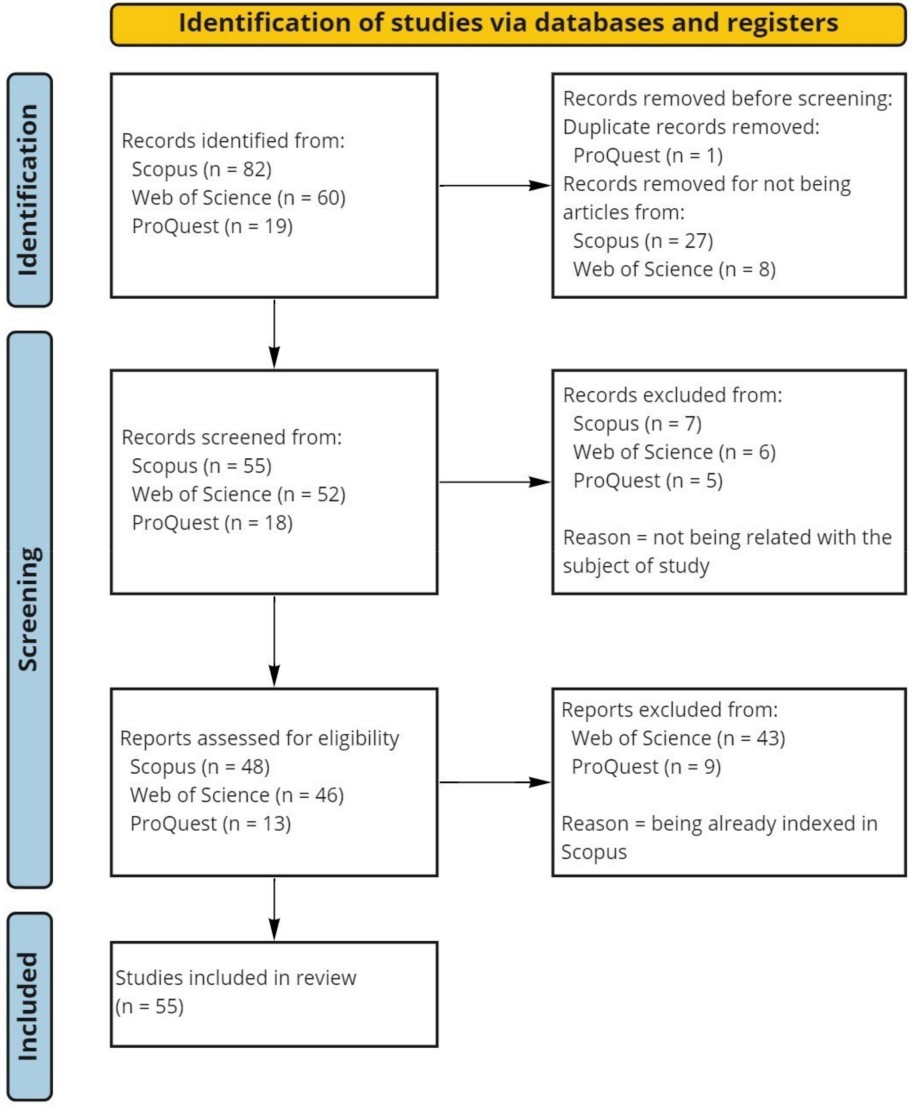

**Figure 3.** PRISMA flow diagram.

## 3. Results

### 3.1. Productivity and Impact Metrics

In Table 1, it is possible to see the number of publications summarized by year. The year with the most publications was 2020 with 14; this means there were 250% more than in 2017. The year 2022 will exceed 14 publications, considering that, to date (10 July 2022), there were already 12 articles published.

**Table 1.** Number of publications on "Solo Travel" by year.

| Publication Year | No. Publications |
| --- | --- |
| 2022 | 12 |
| 2021 | 11 |
| 2020 | 14 |
| 2019 | 6 |
| 2018 | 8 |
| 2017 | 4 |
| Total | 55 |

In Figure 4, it is possible to see the journals where the articles were published. Journals with at least two articles published relating to "Solo Travel" were included. The most striking information that can be obtained from this graph is that two of the articles published were in the journal "Travel Medicine and Infectious Disease"; this has to do with the COVID-19 pandemic, the implications it had on the tourist and travel levels, and the academic studies that were carried out related to this topic. This content analysis will analyze the content of these articles in more detail. The journal with the highest number of publications related to "Solo Travel" was "Current Issues in Tourism" with five, followed by the "International Journal of Tourism Research", and "Tourism Management Perspectives," with four articles each.

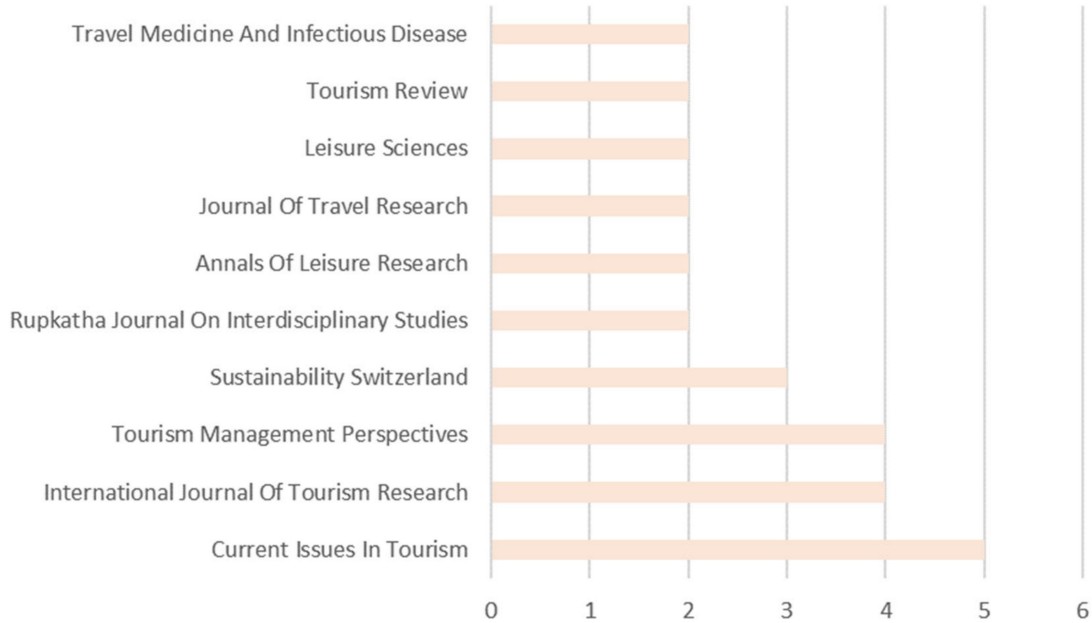

**Figure 4.** Top journals contributing to the subject of "Solo Travel" (at least two articles published).

In Figure 5, it is possible to see universities where the authors were affiliated at the time of publishing the articles; only universities which at least two published articles were included. Most of the authors who published on the subject "Solo Travel" were affiliated with Griffith (Business School and University), followed by Hong Kong Polytechnic University.

Figure 6 is a map that shows the countries with the most articles published on the subject of "Solo Travel", with at least two articles published. Australia is the country with the most published articles (12), followed by the United States (10) and then the United Kingdom, China, and India (5 each).

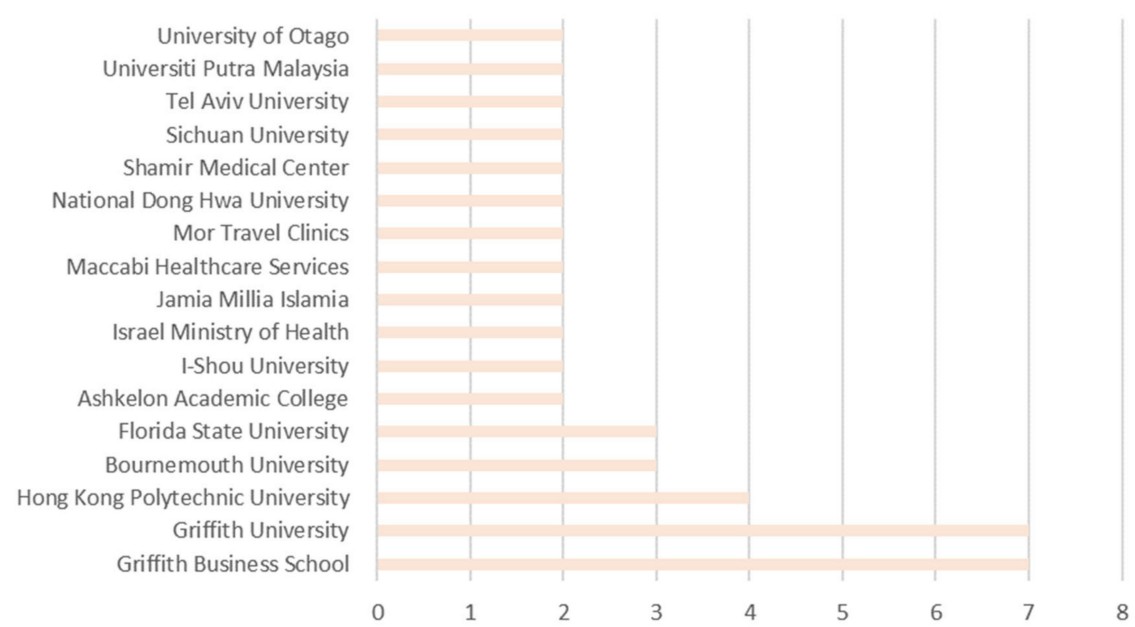

**Figure 5.** Top universities contributing to the subject of "Solo Travel" (at least two articles published).

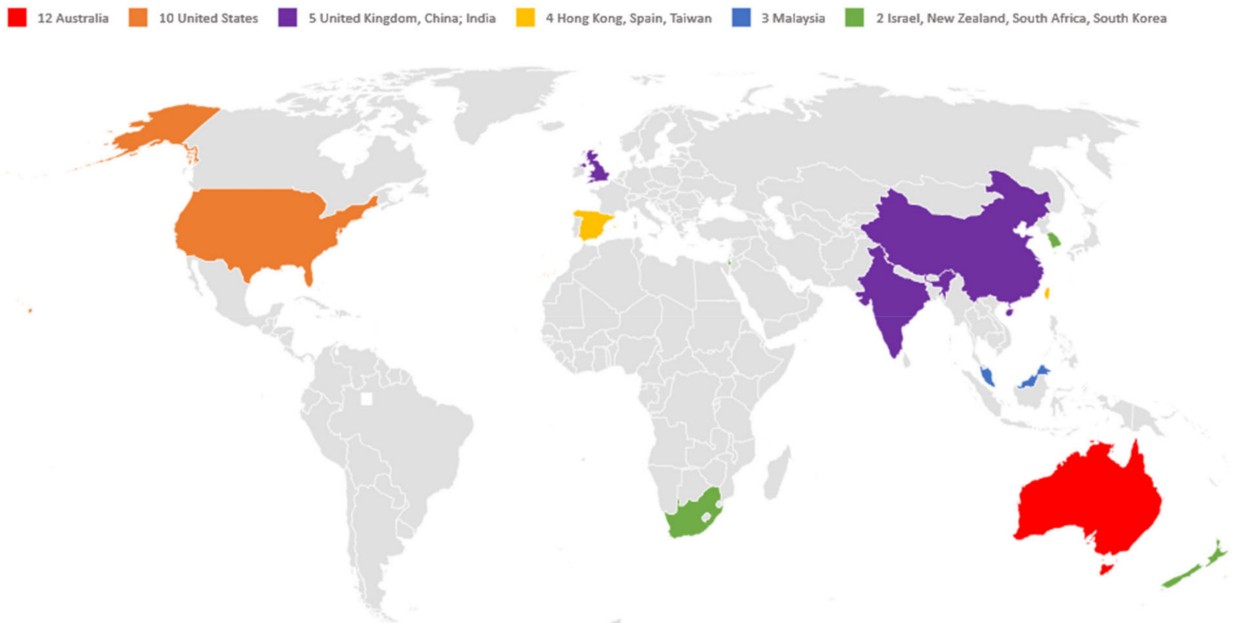

**Figure 6.** Top countries contributing to the subject of "Solo Travel" (at least two articles published).

Figure 7 shows the authors with the most published articles, with at least two published. It is possible to see that three female authors stand out in the number of publications related to "Solo Travel": Yang, Elaine Chiao Ling (six articles); Brown, Lorraine (three articles); and Khoo, Catheryn (three articles).

Table 2 presents 10 of the most cited articles related to solo travel. The article with the most citations (44) is "Power and empowerment: How Asian solo female travellers perceive and negotiate risks", followed by "The solo female Asian tourist" (43) and "Constructing Space and Self through Risk Taking: A Case of Asian Solo Female Travelers" (41), all of them published in 2018. They were published in "Tourism Management, Current Issues in Tourism", and the "Journal of Travel Research", respectively. The tenth most cited article, "Big data analysis of Korean travelers' behavior in the post-COVID-19 era", has been cited only 18 times; this article was published in 2021 by the journal "Sustainability".

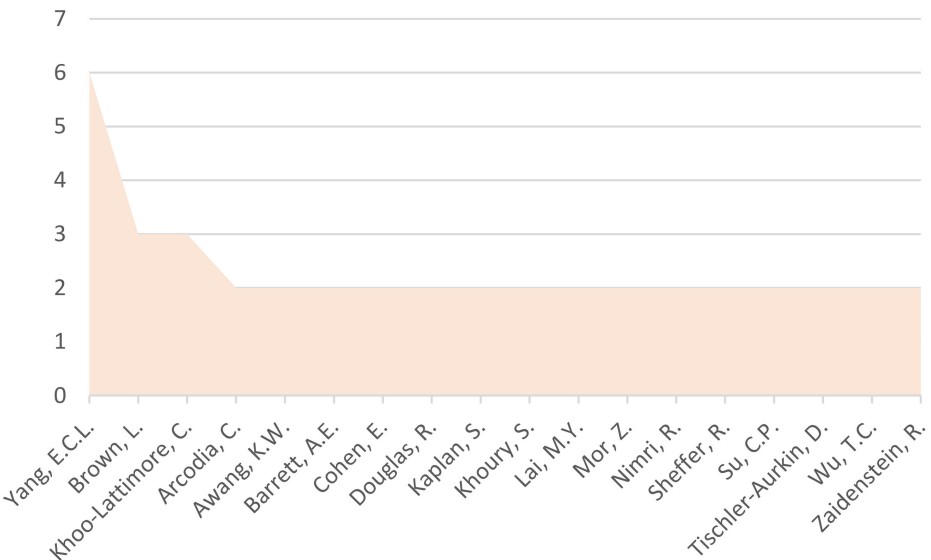

**Figure 7.** Top authors contributing to the subject of "Solo Travel" (at least two articles published).

**Table 2.** Top 10 most cited articles on the subject of "Solo Travel".

| Title | Authors | Publication Year | Journal | Total Citations |
|---|---|---|---|---|
| Power and empowerment: How Asian solo female travellers perceive and negotiate risks | Yang E.C.L., Khoo-Lattimore C., Arcodia C. | 2018 | *Tourism Management* | 44 |
| The solo female Asian tourist | Seow D., Brown L. | 2018 | *Current Issues in Tourism* | 43 |
| Constructing Space and Self through Risk Taking: A Case of Asian Solo Female Travelers | Yang E.C.L., Khoo-Lattimore C., Arcodia C. | 2018 | *Journal of Travel Research* | 41 |
| Tourist satisfaction and subjective well-being: An index approach | Saayman M., Li G., Uysal M., Song H. | 2018 | *International Journal of Tourism Research* | 32 |
| The role of perceived behavioural control in the constraint-negotiation process: the case of solo travel | Chung J.Y., Baik H.-J., Lee C.-K. | 2017 | *Leisure Studies* | 27 |
| How does family influence the travel constraints of solo travelers? Construct specification and scale development | Yang R., Tung V.W.S. | 2018 | *Journal of Travel and Tourism Marketing* | 25 |
| A hybrid method with TOPSIS and machine learning techniques for sustainable development of green hotels considering online reviews | Nilashi M., Mardani A., Liao H., Ahmadi H., Manaf A.A., Almukadi W. | 2019 | *Sustainability* (Switzerland) | 24 |
| The meanings of solo travel for Asian women | Yang E.C.L., Yang M.J.H., Khoo-Lattimore C. | 2019 | *Tourism Review* | 20 |
| The travel motivations and experiences of female Vietnamese solo travellers | Osman H., Brown L., Phung T.M.T. | 2020 | *Tourist Studies* | 19 |
| Big data analysis of Korean travelers' behavior in the post-COVID-19 era | Sung Y.-A., Kim K.-W., Kwon H.-J. | 2021 | *Sustainability* (Switzerland) | 18 |

*3.2. Content Analysis*

Table 3 shows an analysis of the most frequently used words in titles, abstracts, and keywords, excluding determinants, prepositions, adverbs, and pronouns. The terms travelers and travelers were added up and counted only as travelers. In addition to solo, travel, and travelers, other words included: female; women; Asian; or constraints that appear as the most used words in titles, abstracts, and keywords. Other words, such as risk, experiences, motivation, or negotiation, were also commonly used. This information coincides with the key theme timeline and with the content analysis that was performed.

As can be seen in Figure 8, in the evolution of solo travel research, the main topics and themes have not changed since 2017; if not, others were added, thereby expanding our knowledge about solo travel. In this way, women serve as the central theme of solo travel research: constraints, safety, travel behavior, and focus on Asian women are themes that have remained throughout the years. From 2018 onward, other topics began to be added, such as motivations, experiences, risks, the influence of the family when making decisions, or perceptions about leisure. In 2019, the research extended to men and to more specific topics, such as accommodation and destinations, and the topic of identities was increasingly included. In 2020, the field of mobilities continued to expand, and more attention was paid to the segmentation of the solo travel market in research. In addition, research has increasingly considered the digital world and the travel advice generated through it. Comparative studies gained strength in 2021 with comparisons between solo and non-solo travelers, something that was claimed as necessary in earlier investigations. Women's encounters with men were included, not as something negative, but rather as something that can be positive and desired by both women and men. Solo travel narrative took on some importance in solo travel research in 2021. Finally, in 2022, the trend toward comparative studies was confirmed, in this case, with comparisons between women and men. The field of research continues to extend into specific topics, such as museums or the implications of the COVID-19 pandemic in this type of tourism. Of the 55 articles analyzed, 17 are not specific to solo travel, but the term solo travel appears in them as a segment of travelers to make comparisons with other types of travelers, for example, with travelers traveling with family or friends. It is curious that, when the articles are not specific to solo travel, the term is used to make comparisons; however, when the article is specific to solo travel, it is not until recently that comparisons have been made with other types of travelers. To make comparisons between distinct types of travelers is something that was demanded in many earlier investigations to improve and broaden knowledge on solo travelers. Likewise, when the articles are not specific to solo travel, there is no difference between women and men; solo travelers are analyzed as a single block. Those articles not specific to solo travel offer varied knowledge about this tourist segment in terms of quality requirements, complaints, or tourist satisfaction. Of the 38 articles that are specific to solo travel, 25 are studies where only women are included; only 11 include the study of both genders; and only 1 is focused on men, though it also includes women. In total, only 1 of the 38 articles is from a gender comparison perspective. Although there are other studies that include both genders, the article "Reflecting the convergence or divergence of Chinese outbound solo travelers based on the stimulus-organism-response model: A gender comparison perspective" [29] considers gender differences from a gender perspective, not as a simple segmentation between women and men. There are three articles in which the central theme is health, but they supply interesting information about solo travelers. "Airborne Transmission of SARS-CoV-2 Delta Variant within Tightly Monitored Isolation Facility, New Zealand" is a study about the transmission of COVID-19; the article analyzed how a solo traveler can transmit COVID-19 to another tourist in a hotel [35]. "Adherence to antimalarial chemoprophylaxis among Israeli travelers visiting malaria-endemic areas" concluded that solo travelers are less adherent to antimalarial chemoprophylaxis compared with those who travel with a family member and that solo travelers are less prepared for travel than those traveling with a companion [58]. "Morbidity among Israeli backpack travelers to tropical areas" concluded that illnesses/injuries are

more common among females and solo travelers compared with those who travel with their families/partners [34].

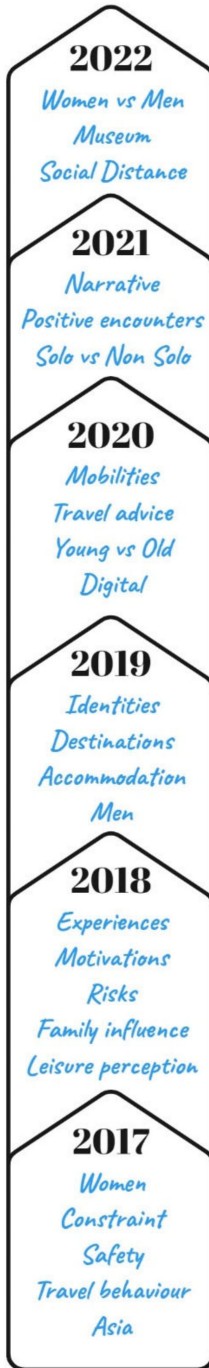

**Figure 8.** Key theme timeline, 2017–2022.

Table 4 is a content analysis of the five most cited articles and the five most recent publications; the five most recent articles were all published in 2022. In the list of most recent articles is the article "Airborne Transmission of SARS-CoV-2 Delta Variant within Tightly Monitored Isolation Facility, New Zealand", but it was excluded from the analysis because it is more health-related, and the content in this article related to solo travel has already been explained above.

**Table 3.** Title/abstract and keyword analysis for "Solo Travel".

| TOP 23 Words on Titles | | | *n* = 638 | TOP 21 Words on Abstracts | | | *n* = 9135 | TOP 21 Keywords | | | *n* = 512 |
|---|---|---|---|---|---|---|---|---|---|---|---|
| Rank | Variable Name | Absolute Frequency | Relative Frequency | Rank | Variable Name | Absolute Frequency | Relative Frequency | Rank | Variable Name | Absolute Frequency | Relative Frequency |
| 1 | solo | 33 | 5.17% | 1 | solo | 183 | 1.91% | 1 | travel | 35 | 6.84% |
| 2 | travel | 20 | 3.13% | 2 | travel | 158 | 1.65% | 2 | solo | 31 | 6.05% |
| 3 | travelers | 17 | 2.66% | 3 | travelers | 82 | 0.86% | 3 | female | 15 | 2.93% |
| 4 | female | 15 | 2.35% | 4 | women | 71 | 0.74% | 4 | tourism | 11 | 2.15% |
| 5 | women | 7 | 1.10% | 5 | study | 68 | 0.71% | 5 | travelers | 11 | 2.15% |
| 6 | experiences | 6 | 0.94% | 6 | female | 63 | 0.66% | 6 | women | 8 | 1.56% |
| 7 | Asian | 5 | 0.78% | 7 | tourism | 53 | 0.55% | 7 | leisure | 7 | 1.37% |
| 8 | tourism | 5 | 0.78% | 8 | were | 49 | 0.51% | 8 | constraints | 7 | 1.37% |
| 9 | online | 4 | 0.63% | 9 | are | 46 | 0.48% | 9 | cultural | 6 | 1.17% |
| 10 | study | 4 | 0.63% | 10 | research | 44 | 0.46% | 10 | Asian | 5 | 0.98% |
| 11 | intentions | 4 | 0.63% | 11 | Asian | 38 | 0.40% | 11 | motivation | 5 | 0.98% |
| 12 | case | 4 | 0.63% | 12 | was | 37 | 0.39% | 12 | gender | 5 | 0.98% |
| 13 | male | 3 | 0.47% | 13 | findings | 35 | 0.37% | 13 | tourist | 5 | 0.98% |
| 14 | tourist | 3 | 0.47% | 14 | constraints | 32 | 0.33% | 14 | risk | 5 | 0.98% |
| 15 | social | 3 | 0.47% | 15 | social | 32 | 0.33% | 15 | negotiation | 4 | 0.78% |
| 16 | role | 3 | 0.47% | 16 | experiences | 29 | 0.30% | 16 | satisfaction | 4 | 0.78% |
| 17 | constraints | 3 | 0.47% | 17 | analysis | 26 | 0.27% | 17 | theory | 4 | 0.78% |
| 18 | traveling | 3 | 0.47% | 18 | have | 26 | 0.27% | 18 | consumer | 4 | 0.78% |
| 19 | practice | 3 | 0.47% | 19 | traveling | 26 | 0.27% | 19 | constraint | 4 | 0.78% |
| 20 | alone | 3 | 0.47% | 20 | has | 26 | 0.27% | 20 | online | 4 | 0.78% |
| 21 | analysis | 3 | 0.47% | 21 | experience | 26 | 0.27% | 21 | social | 4 | 0.78% |
| 22 | negotiation | 3 | 0.47% | | | | | | | | |
| 23 | perspective | 3 | 0.47% | | | | | | | | |

**Table 4.** Content analysis of the most cited and the most recent publications regarding "Solo Travel" (*n* = 10).

| Title<br>Authors | Content |
| --- | --- |
| Power and empowerment: How Asian solo female travellers perceive and negotiate risks<br>Yang E.C.L., Khoo-Lattimore C., Arcodia C. (2018) | The article explores how Asian women perceive and negotiate the risks of traveling alone via constructivist-grounded theory. Results show the concerns of Asian solo female travelers. Results also show individual transformation and empowerment through negotiating risks despite unequal power relations in a gendered and racialized tourism space. |
| The solo female Asian tourist<br>Seow D., Brown L. (2018) | This article, through in-depth interviews, performs a thematic analysis of the travel motivations, experiences, and constraints of solo female Asian tourists. Sexual male attention, harassment, and sociocultural expectations are important constraints for solo female Asian tourists. However, these constraints do not deter these women from solo travel. |
| Constructing Space and Self through Risk Taking: A Case of Asian Solo Female Travelers<br>Yang E.C.L., Khoo-Lattimore C., Arcodia C. (2017) | This article, within a feminist framework, aims to look deeply into the risk perception and risk management of Asian solo female travelers. Moreover, risk and tourist experience are connected; results show how existing tourism spaces remain gendered and Western-dominated and how negotiating risk is also a way to negotiate gender identities. |
| Tourist satisfaction and subjective well-being: An index approach<br>Saayman M., Li G., Uysal M., Song H. (2018) | This article, through a questionnaire focused on tourist satisfaction indices, studies the impact of travel experiences on tourist satisfaction and on their sense of well-being. Results show that the higher the impact of the trip on a tourist's sense of well-being, the higher the loyalty toward the destination. Group travelers had significantly more positive experiences compared with solo travelers. |
| The role of perceived behavioural control in the constraint-negotiation process: the case of solo travel<br>Chung J.Y., Baik H.-J., Lee C.-K. (2017) | This article extends the leisure constraint–effects–mitigation model to the perceived behavioral control (PBC). Results suggest that PBC mediates the relationship between motivation and negotiation and that there is a direct path from motivation to participation. The model was extended to different types of travelers, such as, for example, the case of solo travelers [59]. |
| Reflecting the convergence or divergence of Chinese outbound solo travellers based on the stimulus-organism-response model: A gender comparison perspective<br>Yang, J., Zhang, D., Liu, X., Li, Z., Liang, Y. (2022) | This article, based on the stimulus–organism–response model, focuses on Chinese solo tourists and aims to examine the relationships between cultural distance, emotional solidarity, and perceived safety on tourist behavioral intentions. Results show gender differences between solo travelers in terms of the influence that cultural distance, emotional solidarity, and perceived safety have on their behavioral intentions. |
| Antecedents of tourists' solo travel intentions<br>Bianchi, C. (2021) | This article, via the theory of planned behavior and by incorporating variables such as tourist satisfaction, pleasure, and self-development, aims to investigate the predictors of tourists' intentions to continue solo traveling. An online survey was applied to solo tourists from different countries and of all genders. Results show that, except for subjective norms, all the variables are significant predictors of tourists' intentions to continue solo traveling. |
| Do constraint negotiation and self-construal affect solo travel intention? The case of Australia<br>Yang, E.C.L., Lai, M.Y., Nimri, R. (2022) | This article aims to investigate, through a PLS-SEM model on an Australian sample, the effect of motivations and constraints on solo travel intentions by considering constraint negotiation and the influence of self-construal and PBC. Results show that self-actualization, self-construal, and PBC are key factors in solo travel intention. On the contrary, interpersonal constraints negatively affect solo travel intention. |
| The exploration of Iranian solo female travellers' experiences<br>Hosseini, S., Macias, R.C., Garcia, F.A. (2022) | This article, through in-depth interviews, examines the travel experiences of Iranian solo female travelers. Results reveal that freedom and flexibility, self-empowerment, independence, and exploration are solo travel motivations for Iranian women. At the same time, the absence of family responsibilities, routines, and gender constraints, as well as the promotion of their social and personal selves, contributes to their well-being. |
| The influence of travel companionships on memorable tourism experiences, well-being, and behavioural intentions<br>Vada, S., Prentice, C., Filep, S., King, B. (2022) | This article examines, through a structural equation model in an Australian sample, the role of companionship in memorable tourism experiences, traveler well-being, and behavioral intentions. Results reveal differences in attitudes between those accompanied and those traveling solo. Solo travelers, although traveling alone, show a need to share their experiences with family and friends upon returning from their travels. |

## 4. Discussion

The results of this research confirm the previous literature's findings, in that solo travel is an underexplored area in scientific research, not only in terms of the small number of existing publications, but also the few citations they receive. In addition, research—which previously focused on Western regions, as the previous literature pointed out—is now focused on the Asian region. Moreover, also in line with the previous literature, the results of this research show that most of the studies are focused on women. Few of them are presented from a comparative perspective between genders, between segments within the solo tourist, or in comparison with other types of tourists. There is also a concentration of authors who are experts on this topic (all of them are female authors) and of universities to which these authors are affiliated. It is necessary to increase the scientific literature on solo travel not only because of the growing impact of solo travel on the tourist sector, but also because of its implications for gender equality and because of *the "lack of exposure and education on solo women's travel that could play a role in perpetuating the cycle of imposed social norms"* [27]. There are few studies referring to more countries or regions, comparing all genders, or comparing with other segments. In addition, it is necessary to extend the topics of research beyond constraints, experience, risks, motivations, etc... It would also be interesting to have more male authors writing about solo travel from a gender perspective.

Perhaps, at this point, the most important thing is to reach a consensus on what factors constitute solo travelers; this is essential because, otherwise, we might be studying different tourist groups, making analysis more difficult and even leading to contradictory results, as seen in the presented literature. The authors of this article think that the correct decision is to consider solo travelers to be those people who arrive alone at a destination, no matter whether they join a group or a tour or if they finish the trip with someone they meet during the trip, provided that there are no friends or family. In the end, even when a solo traveler joins a group, tour, or meets people along the way, she or he will have to deal with many situations alone. The authors also believe that business travelers, who have different motivations, experiences, constraints, and behaviors, should not be included in the segment of solo travelers. As seen in the previous literature, Laesser et al., (2009) establish four classes of single travelers: people living alone who travel alone; people living alone who join a group or an organized trip; people who do not live alone and travel alone; people who do not live alone and travel alone but join a group or an organized trip. Determining whether these people live alone or not is important since this distinction could be the cause of many decisions taken when they arrive at the destination. For example, people who live alone may take longer trips or may be able to afford to spend more money than people who do not live alone. This could be an interesting topic for future research. What seems clear is that excluding people who travel alone but join a group or an organized tour from the term solo traveler is something that is far from reality, since people who travel alone often do so in organized packages [5]. In fact, as stated by tour package organizers, the rates of single travelers who book a tour have not stopped rising; some of those companies calculate that they are receiving 300% more bookings from people traveling alone than from people traveling as couples or with family or friends [9]. The authors, therefore, consider it necessary to reach a consensus on what a solo traveler is, and the distinction made by Laesser et al., (2009) seems to be the most appropriate so far. Tourists should be considered solo travelers if they arrive alone at a destination, no matter whether they finish their travels alone or are accompanied by unknown people, either because they joined a group or tour or because they met people during the journey. This definition should also differentiate between those who live alone and those who do not and between those who travel alone because they want and those who have no one to travel with; not having someone to travel with is compatible with living with someone else, for example, having holidays on different dates from those of the other member of the couple. As such, the authors propose to add people who travel alone "by default" or "by choice" to Figure 1 in Laesser et al., (2009, p. 161). In this way, solo travelers could be classified as per Table 5.

**Table 5.** Solo traveler classification.

| Type of Household Traveler Comes from | Type of Network Traveler Travels in | | Reason to Travel Alone |
| --- | --- | --- | --- |
| | *Solo* | *Group or Tour of People Previously Unknown (More Than One)* | |
| *Single (one person only)* | SINGLE—SOLO—DEFAULT | SINGLE—GROUP—DEFAULT | *By default* |
| *Collective (more than one person)* | COLLECTIVE—SOLO—CHOICE | COLLECTIVE—GROUP—CHOICE | *By choice* |

The authors consider this distinction important and believe that it would modify many of the results obtained so far in solo travel studies, for example, by establishing self-knowledge, quest for solitude, making one's own decisions, empowerment, etc. as motivations. However, these motivations might not be valid for those traveling alone because they have no one to travel with. People who have no one to travel with might have the same motivations as those traveling accompanied, but they travel alone because they have no choice. Therefore, it is also important to perform segmentations within solo travelers by age, sex, country of origin, religion, the type of tourism they will practice, etc., as well as comparisons with other types of tourists. Studies cannot consider solo travelers a homogeneous group, and when segmentation and comparison start to be used more regularly in solo travel research, the motivations, experiences, constraints, etc. will be much more varied and different from those presented so far. This expansion of topics related to solo travel, along with the need to segment and compare, has practical implications in the tourism sector because it would allow to know where to target marketing policies, campaigns, and strategies aimed at solo travelers. The current knowledge leads to campaigns focusing on experience, overcoming, or disconnecting within a friendly and safe environment. However, the research published so far does not make it possible to ascertain whether men and women have different motivations, or whether these motivations are different from those of people who do not travel alone, nor it is possible to know whether these motivations are different depending on the countries of origin and countries of destination of travelers. The second implication is related to safety; policy managers, together with tourism managers, must take care in creating safe places, especially for women who travel alone, as well as tools or facilities to feel safe once in a destination. In this case, it is also important to increase the knowledge of dangers and constraints faced by people who travel alone and to analyze if these dangers and constraints are different from people who travel accompanied. It is also important to know differences according to sex, country of origin, and destination. This improvement of knowledge would not only lead to an increase in solo travel tourism—especially considering that first-time solo travelers, if they enjoy the experience, usually do so again—but it will also contribute to achieving Goal 5 (gender equality) of the 2030 Agenda. Therefore, expanding the field of study does not mean that the gender perspective should be excluded in further solo travel studies. On the contrary, the solo travel research published so far has made a huge contribution to gender equality.

Publications should continue to include the analysis of gender factors associated with this type of tourism, such as safety, sexual harassment, or its contribution to gender equality. It should not be analyzed as something isolated from tourism, but rather, as part of the problems and risks that women face in their daily lives. Women's bodies often invite uncomfortable attention and sexual advances [36]; women suffer stereotypes, discrimination, and inequality, which are maximized by the religious and cultural environments of each country. However, the truth is that women suffer the same constraints in their daily lives without having to go on a trip to be afraid, harassed, or sexually attacked. Women are afraid to go out alone at night, even in the places where they live. They choose the itinerary to return home to make it safer, and they change clothes according to the places they are going to because of the discomfort caused by male attention; families may be afraid that

they will go out alone and even persuade them not to go. These risks cannot be treated as if they were something specific to tourism, as they are the daily risks and constraints of millions of women in the world. Thus, these problems should not be treated in the academy as if they were something exclusive to women who travel. On the other hand, it is important to focus on the aggressors and not so much on women; it is important to focus on those responsible for women having to live and travel in fear, as would it is for any type of violent or criminal act. The literature speaks of women as *voluntary risk-taking adventurous travelers* [29,30] or indicates that *the concern of safety and security . . . once they conquer this feeling . . . becomes a reward for the journey* [8]. This could place the focus on women and *place the responsibility on women to be aware of their potentially problematic status as a woman alone* [26]. Women keep traveling because, if they stopped doing things out of fear, they could not do anything in their lives. Thomas and Mura (2019) stated that *solo female travelers have internalized the normality of unsafety*, and it is completely true; women have internalized the normality of insecurity, not only in travel, but in all aspects of their lives. To say that overcoming those fears and insecurities is a form of empowerment is another trap of male hegemony. A sexual attack should not be considered one more risk of traveling; continuing to travel despite running the risk of being sexually assaulted should not be considered a form of empowerment. It is necessary to research ways to make solo female travel safer without placing the responsibility for maintaining their own safety on women. In our opinion, it is the responsibility of the academy to continue asking women if they feel safe or if insecurity prevents them from traveling, with the purpose of seeking possible solutions so that they can feel and be safe, such as in, for example, the article *Research and Design for Hotel Security Experience for Women Traveling Alone* (2020), where after finding out where women feel unsafe inside a hotel, the authors looked for ways to increase safety in those places.

This article was presented as a critical review to propose other research paths related to solo travel. The authors are aware of the difficulty of this topic because of its implications for gender equality, and this article is not without limitations. The main limitation of this study is the selection of words for filtering; authors used the words "Solo near/1 travel*" to allow to find phrases such as "Solo travelers", "Solo female travel", "Solo men travel", and so on. However, for example, words such as "travel alone" could also be used. On the other hand, although the selected databases are the most complete databases, with the largest number of scientific and academic publications, there may be other articles not indexed in these databases that may have been ignored. At the same time, in all the manuscript screening during the first search, those that were not articles were excluded from the analysis. Performing a search without excluding those manuscripts, and increasing the number of databases used, could increase the number of analyzed manuscripts and, therefore, the information obtained.

## 5. Conclusions

Research on solo travel is complex for two reasons. On the one hand, this is because solo travelers are a very heterogeneous type of tourist, as evidenced by the fact that the literature published so far has not reached a consensus on the definition of a solo traveler. On the other hand, this is also because the study of solo travel implies the need to do so from a gender perspective due to the added risks that this type of tourism entails for women, a fact that was noted throughout this article. Research focused on solo travel should increase; new investigations should segment solo travelers (sex, age, type of tourism they practice, religion, country of origin, etc.) and realize comparisons between those segments and other tourists who travel accompanied. To do so, establishing a consensual definition of solo travelers is necessary. In addition, research should extend to other regions beyond Asia and expand the field of analysis beyond motivations, experiences, or constraints. Research focused on solo female travelers should continue, not only because of the growing impact of solo travel on the tourist sector, but also because as long as women cannot travel under the same conditions as men, effective gender equality cannot be achieved. New research

must look into the causes of inequality and the causes of insecurity suffered by women, not to make women responsible for their own security, but to look for solutions so that they can feel and be safe.

**Author Contributions:** Conceptualization, A.O.-C. and N.A.-V.; methodology, N.A.-V. and A.O.-C.; validation, A.O.-C.; formal analysis, A.O.-C.; investigation, A.O.-C.; data curation, A.O.-C.; writing—original draft preparation, A.O.-C.; writing—review and editing, A.O.-C.; visualization, A.O.-C.; supervision, N.A.-V. and J.A.F.-B.; project administration, N.A.-V. and A.O.-C. All authors have read and agreed to the published version of the manuscript.

**Funding:** This research received no external funding.

**Institutional Review Board Statement:** Not applicable.

**Conflicts of Interest:** The authors declare no conflict of interest.

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
