# Peer review of "Solo Travel Research and Its Gender Perspective: A Critical Bibliometric Review"

_tourismhosp, doi:10.3390/tourhosp3030045_

Round 1
Reviewer 1 Report
First, I congratulate the authors for their work and I thank them for the opportunity they have given me to read this paper.
I congratulate you for the methodology used.
The work is clear, well structured, with an interesting discussion that can add new knowledge to the academy.
However, when we are invited to review an academic text, I think it is a very important commitment that deserves a very deep look. That said, I have very few observations that can be taken as recommendations. But I leave it up to the editor to make the decision whether or not to recommend these changes. All this because the text is very complete and very well written.
1. After reading it, it seems to me that the article talks more about Asian single tourism. I don't know if it's my perception, but I get the impression that the bibliographical review mainly identified papers of authorship/theme of Asians.
Wouldn't that be a work limitation? Perhaps, wouldn't a conclusion on this be interesting? Could it be that it is not worth expanding the discussion a bit towards this path? In other words, although they have not looked for this specific audience, in this article it became clear that single tourism goes hand in hand with Asian women. And I'm sure that's not the case. In Brazil, Portugal and Spain there is a lot of publication on the topic "women traveling alone".
2. I think that the conclusions detract from the work done. What are the single+woman implications for the industry? Can you make any recommendations regarding research in this field? Not only that, can't practical implications be drawn from the study?
I think there are only two details that deserve to be reviewed so that the paper is as big as it deserves.
Congratulations on the investigation.
Thank you Antic for the invitation to review this work...
Reviewer 2 Report
The introduction needs to be re-visited further reflecting the key selling points of this study, and its potential to contribute to theory. Authors need to emphasize on those points at the outset of the paper. I am aware of the explanations in the introduction (discussion of the definition of solo travel, gender and specific cultural focuses, etc.), but all these are presented in a somewhat scattered way.
The male-female discussion within the scope of solo travel is good. However, the authors continue this discussion without emphasizing the concept of "meaningful travel", especially for female solo travelers. I think the phenomenon of meaningful travel must be included.
Some references are not in the right place or are missing. For example; "family and social customs" on page 4 is Karagoz et al.'s direct research finding, but this reference is missing.
The time interval of the research was determined as 2017-2021. However, the authors did not explain why they determined such a limited time interval in the method section. I do not think that the knowledge revealed due to such a limited time interval is also limited.
